# Transgenic Rice Plants Expressing Artificial miRNA Targeting the Rice Stripe Virus *MP* Gene Are Highly Resistant to the Virus

**DOI:** 10.3390/biology11020332

**Published:** 2022-02-19

**Authors:** Liya Zhou, Quan Yuan, Xuhong Ai, Jianping Chen, Yuwen Lu, Fei Yan

**Affiliations:** 1State Key Laboratory for Managing Biotic and Chemical Threats to the Quality and Safety of Agro-Products, Key Laboratory of Biotechnology in Plant Protection of Ministry of Agriculture and Zhejiang Province, Institute of Plant Virology, Ningbo University, Ningbo 315211, China; liyazhou202202@163.com (L.Z.); tgcf13313983510@163.com (X.A.); jianpingchen@nbu.edu.cn (J.C.); 2College of Plant Protection, Northwest A & F University, Yangling 712100, China; yuanquan0405@126.com

**Keywords:** rice stripe disease, amiRNAs, movement protein, marker-free transgenic rice

## Abstract

**Simple Summary:**

Rice stripe virus is a disastrous viral disease that causes significant yield losses in rice production in South, Southeast, and East Asian countries. To decrease the use of chemical insecticides, genetic engineering has become a pivotal strategy to combat the virus. In this study, we constructed a dimeric artificial microRNA precursor expression vector that targets the viral MP gene based on the structure of the rice osa-MIR528 precursor. Marker-free transgenic plants successfully expressing the MP amiRNAs were obtained and were highly resistant to RSV infection. The novel rice germplasms generated are promising for RSV control.

**Abstract:**

Rice stripe virus (RSV) causes one of the most serious viral diseases of rice. RNA interference is one of the most efficient ways to control viral disease. In this study, we constructed an amiRNA targeting the RSV MP gene (amiR MP) based on the backbone sequence of the osa-MIR528 precursor, and obtained marker-free transgenic rice plants constitutively expressing amiR MP by *Agrobacterium tumefaciens*-mediated transformation. A transient expression assay demonstrated that dimeric amiR MP could be effectively recognized and cleaved at the target MP gene in plants. Northern blot of miRNA indicated that amiR MP-mediated viral resistance could be stably inherited. The transgenic rice plants were highly resistant to RSV (73–90%). Our research provides novel rice germplasm for RSV control.

## 1. Introduction

Rice stripe disease, caused by rice stripe virus (RSV), is one of the most devastating viral diseases in rice production worldwide. The viral disease first appeared in Japan, and then emerged in over 20 provinces in China, including Jiangsu and Zhejiang. In the period 2002 to 2004, an outbreak of RSV caused devastating economic losses and affected about 80% of the rice cultivated in Jiangsu province [1,2]. RSV is classified in the genus *Tenuivirus,* and its genome has four single-stranded RNA genome segments. RNA1 encodes the RNA-dependent RNA polymerase (RdRp) on the complementary strand, while RNAs 2, 3 and 4 have an ambisense coding strategy [3]. RNA4 encodes a disease-specific protein (SP) in the virion sense and a movement protein (MP) in the complementary sense [4,5]. RSV is transmitted in a persistent, circulative–propagative manner by the planthopper *Laodelphax striatellus*, which makes it difficult for integrative management [6]. The deployment of resistant rice varieties is considered to be the most economic and efficient way to control the disease [7]. Traditional breeding for resistance is slow and inefficient and is challenged by the genetic linkage of agronomic traits with disease resistance genes [8].

In the last few decades, researchers have shown an increasing interest in antiviral genetic engineering in rice. In 1992, an attempt was made to engineer RSV-resistant rice by overexpressing the cp protein. The resulting transgenic plants showed partial RSV resistance but did not generate agriculturally useful lines [9]. In 2011, Takumi et al. designed seven specific siRNAs to target the dsRNAs of each gene in the RSV genome [10]. Rice plants expressing the siRNAs of pc1, CP and MP had the greatest virus resistance.

Among the various transgenic approaches to the creation of virus-resistant plants, one of the most effective involves the expression of artificial miRNA (amiRNA) targeting the virus [11,12]. MicroRNAs (miRNAs) are a class of endogenous, single-strand noncoding RNAs 18–25 nucleotides long, which play important roles in plant defense [13,14]. In 2006, Niu et al. designed new amiRNAs in *Arabidopsis*, which provided resistance to both turnip yellow mosaic virus (TYMV) and turnip mosaic virus (TuMV) [11]. Similarly, an amiRNA targeting cucumber green mottle mosaic virus (CGMMV) protected *Nicotiana benthamiana* from invading viruses and virusoids through RNA silencing mediated by the amiRNA [15]. Antiviral strategies using amiRNA have been successfully used to protect plants against plum pox virus (PPV), potato virus Y (PVY) and cotton leaf curl Burewala virus (CLCuBuV) [16,17,18], while transgenic rice expressing an amiRNA targeting the 3‘-UTR regions of the RSV CP gene had 54% resistance to RSV infection [19]. Compared with the siRNA antiviral strategy, the amiRNA approach overcomes the problem of self-silencing, as miRNAs are not associated with transgene silencing, but also decreases off-target effects [12,20].

In this study, we constructed an amiRNA targeting the RSV MP gene (amiR MP) and obtained marker-free transgenic rice expressing amiR MP. The transgenic plants were highly resistant to RSV and have provided a useful rice germplasm to control RSV.

## 2. Materials and Methods

### 2.1. Sequence and amiRNA Prediction

The amiRNA targeting the segment of the RSV MP gene was identified by the microRNA Target Finder software WMD (http://wmd3.weigelworld.org, accessed on 23 September 2021) and designated as amiR MP [21]. The entire sequence of osa-MIR528 was cloned from rice DNA by PCR amplification using primers primer-F1 and primer-R1 (Appendix A). The predicted amiRNA sequence of RSV MP (TTTCTGAACTACATTAGTCGT) was used to replace the natural miR528/miR528* sequences by oligonucleotide-directed mutagenesis.

### 2.2. Transformation

The *Agrobacterium tumefaciens* strain EHA105 was transformed with the constructs expressing pCV amiR MP and full-length RSV MP by electroporation, and a final OD ratio of 2:1 was used for coinoculation [22]. *N. benthamiana* plants at 3 weeks old were infiltrated using pressure.

### 2.3. Production of T_0_ Generation Transgenic Plants

Rice variety *Oryza sativa* L. var. Zhegeng-88, a commercial rice variety grown in Zhejiang (https://www.ricedata.cn/variety/varis/609074.htm, accessed on 4 February 2022), was used as baseline to generate the transgenic plants. Plants were transformed using *Agrobacterium* as previously described [23] and regenerated on media containing kanamycin sulfate and hygromycin B. To identify transgenic plants, genomic DNA was isolated from leaf tissue by the CTAB method and was screened by PCR with the primers 35S and NOS (Appendix A).

### 2.4. Detection of Foreign Gene Insertion Sites and Homozygotes

The T-DNA insertion sites of T_1_ transgenic lines were determined by thermal asymmetric interlaced polymerase chain reaction (TAIL-PCR) [24]. Primers were designed according to the sequence between the target fragments of LB and RB (Appendix A).

### 2.5. Assessment of Resistance to RSV

Rice plants were inoculated using 10 viruliferous insects (second or third instar *L. striatellus*) per plant and kept in a cage containing 30 plants, as described previously [10]. The numbers of rice plants showing each of the different symptom types on their newly developing leaves were recorded at 4 weeks, as previously described [25].

### 2.6. RNA Analysis

Total RNA was isolated from samples using TRIzol reagent (Invitrogen, Carlsbad, USA) following the manufacturer’s protocol. For Northern blot analyses, 10 μg total RNA was heat-treated at 65 ℃ for 10 min in formamide buffer. Hybridization was performed as described previously [22]. The MP probes were amplified with primers NB-MP-F1/R1 and NB-MP-F2/R2 (Appendix A) and labeled with digoxigenin (DIG).

The miRNA was extracted with an Easy Pure miRNA Kit (TransGen, Beijing, China) according to the manufacturer’s instructions. Wild-type (WT) rice was used as the negative control. For small RNA Northern blotting, approximately 1μg miRNAs were heat-treated at 65 ℃ for 10 min in formamide buffer. miRNAs were separated using a 12% polyacrylamide gel containing 7.5 M urea, transferred electrophoretically to Amersham Hybond^TM^ -N^+^ membranes (GE Healthcare, Hatfield, UK) using H_2_O buffer for 90 min, 0.3 A constant V, and fixed to the membrane by chemical crosslinking. The miRNA probe was commercially labeled with biotin (Sangon Biotech, Shanghai, China). Prehybridization and hybridization used ULTRAhyb^®^-Oligo Hybridization Buffer (Thermo Fisher Scientific, Waltham, MA, USA), and signal detection was performed with Chemiluminescent Nucleic Acid Detection Module (Thermo Scientific, Waltham, MA, USA) according the instruction manual [26].

### 2.7. Real-Time Quantitative PCR and Semiquantitative RT-PCR Analysis

The cDNA was synthesized according to the manufacturer’s protocol of TransScript® All-in-One First-Strand cDNA Synthesis SuperMix for qPCR (One-Step gDNA Removal) Kit (TransGen, Beijing, China). The gene fragments were amplified using 2 x TOROGreen® HRM qPCR Master Mix (TOROIVD Technology, Shanghai, China) and the MP-specific primer for RT-qPCR analysis is listed in Appendix A. The *N. benthamiana* Ubiquitin C (UBC) and *O. sativa* actin genes were used for internal reference control. The ABI QuantStudio5 Real-Time PCR System (Thermo Scientific, Waltham, USA) was used for the reactions and the results were calculated according to the ΔΔCT method. Semiquantitative RT-PCR was used to measure the expression of CP using the primers listed in Appendix A.

### 2.8. Western Blotting

Total proteins for Western blot assay were extracted from leaf samples as described before [22]. Proteins were detected using anti-MP (1:5000) monoclonal primary antibodies and Horseradish Peroxidase-conjugated anti-rabbit secondary antibody (1:10,000; TransGen, Beijing, China). The antigen–antibody complexes were visualized using ECL buffer (Sigma-Aldrich, Saint Louis, MO, USA).

## 3. Results

### 3.1. Construction of the Vector Expressing amiRNA Targeting RSV MP mRNA (amiR MP)

Artificial microRNAs of RSV MP (amiR MP) were predicted and chosen on the website (Figure 1A) and osa-MIR528 precursor was used as the backbone for expressing amiR MP, as described before [12,21]. The natural miR528/miR528* sequences were replaced by viral amiR MP/amiR MP* sequences using oligonucleotide-directed mutagenesis and a precursor of amiR MP (pre-amiR MP) was obtained, as shown in the schematic diagram (Figure 1B). The pre-amiRNA fragment was then inserted into the vector pCV GFP-N2 to obtain transient expression vectors pCV pre-amiR MP (Figure 1B).

### 3.2. Expression of amiR MP Decreased Accumulation of MP mRNA in a Transient Expression Assay

To test whether expression of MP amiRNA precisely cleaves the target mRNA MP, we performed a transient expression assay in *N. benthamiana*. The MP^RSV^ fragment and a synonymous mutant named MP^RSV^SV, which could not be targeted by amiR MP, were inserted into the transient expression vector pCAMBIA1300. As shown in Figure 2A, the green fluorescence of the MP^RSV^SV-GFP and MP^RSV^-GFP was similar without transient expression of amiR MP. However, the intensity of MP^RSV^-GFP fluorescence was sharply decreased when coexpressed with amiR MP (Figure 2A). Western blot using an anti-MP antibody and Northern blot using an *MP* probe confirmed a significant reduction in MP accumulation in the infiltrated patch, but not with MP^RSV^SV (Figure 2B,C). In order to eliminate any effect of miRNA targeting GFP, unfused MP^RSV^SV and MP^RSV^ were transiently expressed. Western blot results showed similar levels of MP^RSV^ and MP^RSV^SV protein accumulation in the absence of amiR MP, but when amiR MP was coexpressed, the transcription and protein level of MP^RSV^ was significantly lower than that of MP^RSV^SV (Figure 2D,E). RT-qPCR confirmed that transcripts of the MP gene in the patch coexpressing MP^RSV^ and amiR MP were at about half the levels of those in the control *MP*^RSV^SV patch (Figure 2F). In summary, the results indicate that the amiRNA of MP effectively recognized and cleaved the target RNA in plants.

### 3.3. Generation of Marker-Free Transgenic Rice Expressing amiR MP

Marker-free transgenic rice plants expressing amiR MP were generated using the cotransformation strategy shown in the schematic diagram (Figure 3A,B). The marker vector pCAMBIA 1300UR-RFP uses pCAMBIA 1300 as a backbone and this contains the maize ubiquitin promoter (ubi-promoter) and rbcS terminator for expression of RFP [27] (Figure 3A). To construct the marker-free expression vector (PCV-HYG-free amiR MP), pCAMBIA1300 was modified by digesting with Ase I to remove the hygromycin B (HYG) resistance gene, but retaining the 35S promoter and a NOS terminator (pCV-HYG-free) (Figure 3A). The prepared rice calli were then subjected to *Agrobacterium tumefaciens*-mediated cotransformation with PCV-HYG-free amiR MP vector and marker vector pCAMBIA 1300UR-RFP. T_0_ generation plants were tested for HYG tolerance selection and PCR detection. Self-fertilization of T_0_ plants allowed the un-linked genes amiR MP and marker RFP to be separated in the T_1_ generation plants. A simple test for RFP expression eliminated the progeny containing the marker gene, and a combination of PCR testing of the remaining plants efficiently identified the marker-free transgenic plants. Seven lines of T_2_-generation marker-free transgenic plants were obtained by this procedure (Appendix A).

To investigate whether the transgenic plants generate amiRNA, we isolated small RNAs from transgenic plants, with WT plants as controls. In Northern blot assays of small RNAs, amiR MP was detected in transgenic plants (lines 18, 19 and 24), but not in the WT (Figure 3C), confirming that expression of amiR MP in transgenic plants was stably inherited.

To eliminate any deviation of miRNA function caused by insertion into a functional region of the genome, we identified the insertion sites of transgenic lines by TAIL-PCR. The amiR MP in line 19 was shown to be located in the gene intergenic spacer of LOC_Os10g4348751 of chromosome 10, and the amiR MP in line 24 between genes LOC_Os10g048840 and LOC_Os10g048800 (Figure 3D). The primers to identify homozygous plants were designed by using the sequences forward and backward of the insertion site and that of T-DNA. When amplified by PCR using the P2/P3 or P2/P5 primers, there was a band from transgenic plants lines 19 and 24 but not from the WT, indicating that the T-DNA had inserted into the rice genome. When amplification was conducted using the P1/P3 primers, the same band was identified in line 19 as in the WT showing line 19 to be heterozygous. Similarly, when amplification was performed using the P4/P5 primers, the band amplified from line 24 was different to that from the WT showing that this line is homozygous (Figure 3E).

### 3.4. Transgenic Rice Plants Were Highly Resistant to RSV

As shown in Appendix A and Figure 4A,B, the phenotypes of the transgenic and wild-type plants were similar. The expression of amiR MP in vivo had little effect on the growth phenotypes such as plant height, tiller number and rate of seed-setting.

To examine if the designed amiRNAs targeting the RSV MP gene conferred resistance in the T_2_-generation transgenic plants, seven transgenic lines (30 plants in each line) were screened and inoculated with RSV by allowing viruliferous vector insects (*L. striatellus*) to feed on them. Wild-type plants were used as controls and the appearance of symptoms was observed daily. After 4 weeks, the symptoms of RSV were recorded on each plant as dead (D), susceptible (S), resistant (R), or immune (I) (Figure 4C). RT-PCR and RT-qPCR were used to detect the accumulation of RSV CP on the inoculated plants (Figure 4D,E). The transgenic plants showed varying degrees of RSV resistance (Table 1) and those with a resistance ratio > 85% (amiR MP-18, amiR MP-19 and amiR MP-24) were considered highly resistant.

## 4. Discussion

Rice production is always constrained by viruses, among which RSV is a devastating pathogen that causes significant yield losses in most Southeast Asian countries. The application of insecticides against the intermediate insect vectors is one of the most successful ways to prevent damage and yield loss from rice viruses, but the high cost and environment pollution are major burdens in rice production [28,29]. With the development of genetic engineering, RNA interference has become a pivotal strategy to combat these viruses [30]. Several studies have introduced siRNAs to target the virus genomes and have obtained transgenic rice resistant to viruses such as rice dwarf virus (RDV) [31,32], RSV [10] and rice black streaked dwarf virus (RBSDV) [33]. Unlike siRNA, miRNAs are single-strand noncoding RNA around 22 nucleotides long [34,35] and there is mounting evidence that they play a critical role in virus infection [36,37]. However, the amiRNA approach has rarely been used in rice to provide resistance to RSV. Here, by using amiRNA-mediated RNA silencing, we generated transgenic plants carrying a dimeric amiRNA precursor producing miRNA targeting the viral MP to resist RSV infection.

A previous study suggested that the variable levels of resistance to viral infection of plants that harbor RNAi constructs depended on the particular viral gene used as the target for RNAi, and that best resistance to RSV was obtained targeting the cp and mp sequences [10]. The T_2_-generation of cp target transgenic rice plants still exhibited immunity to RSV infection. However, unlike siRNA, Sun et al. obtained only 54.17% resistance efficiency in transgenic rice with amiRNA targeting RSV *CP* [19]. In this study, transgenic rice plant lines targeting the RSV MP exhibited a much higher level of resistance to RSV (93%), underlining the importance of selecting the best target sequence.

If there is to be large-scale commercial cultivation of such transgenic crops, plants without selection markers are desired. Such plants are currently usually produced by a cotransformation strategy. Some researchers construct a binary vector that carries two separate T-DNAs, one containing a selection gene and the other for expression of the desired gene [38,39,40]. However, this method is hampered by the difficulty in obtaining cotransformation vectors due to the lack of convenient restriction sites. Another method is to construct two vectors containing different T-DNAs and to transfer them to Agrobacterium separately [41,42]. However, as a cotransformation strategy, this involves a very heavy workload to screen large numbers of transgenic plants by a PCR assay. In 2005, Chen et al. constructed a double T-DNA vector containing the green fluorescent protein (GFP) gene to easily monitor the segregation of marker genes, thus facilitating screening of marker-free progeny [43]. Here, we constructed a marker vector carrying an RFP selection gene for the convenient observation of red fluorescence in rice leaves. Therefore, when we cotransformed with the RFP marker vector and amiR MP vector, we could observe red fluorescence quickly in T_1_-generation transgenic rice under a confocal microscope and use PCR testing to identify the marker-free transgenic plants conveniently. Among the cotransformed plants, we obtained seven lines containing two unlinked T-DNAs, enabling the segregation of transgenic rice in the subsequent generation.

Our marker-gene-free transgenic rice plants contain only the antiviral miRNA and exhibit 90% resistance to RSV infection. As a large number of virus-resistant transgenic crops have emerged, the biological safety of transgenic plants has raised concerns, including off-target effects [44,45]. However, since the transgenic plants express only a short sequence through amiRNA-mediated strategies, these potential risks are likely to be minimal [11]. These plants are a new germplasm resource for breeding rice resistant to RSV.

## 5. Conclusions

In this study, a construct of artificial microRNA of RSV MP based on the backbone sequence of the osa-MIR528 precursor was cotransformed with an RFP marker vector into rice to obtain marker-free transgenic plants. These transgenic plants, constitutively expressing amiR MP, had high resistance to RSV infection.

## Figures and Tables

**Figure 1 biology-11-00332-f001:**
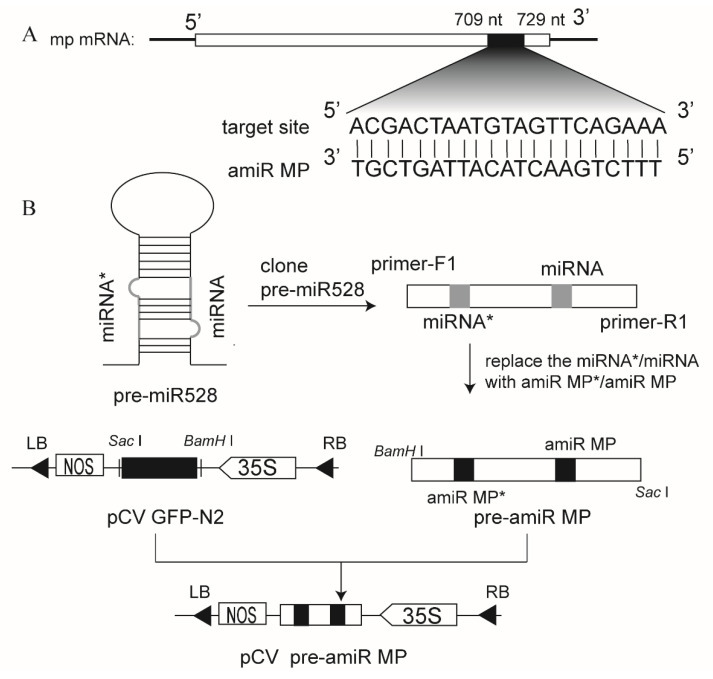
Schematic diagram showing the construction of the expression vector of amiR MP. (**A**) The target site of amiR MP. (**B**) Strategy for artificial miRNA construction. The natural miR528/miR528* sequences were replaced by viral amiR MP/amiR MP* sequences using oligonucleotide-directed mutagenesis and were cloned into the vector pCV GFP-N2. The 35S promoter and Nos-T nopaline synthetase terminator were used in this construction.

**Figure 2 biology-11-00332-f002:**
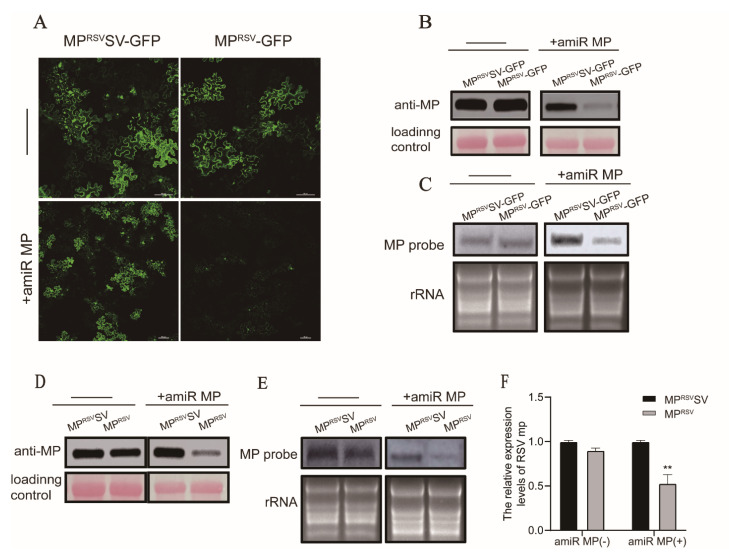
Expression of amiR MP decreased accumulation of MP mRNA in a transient expression assay. (**A**) The green fluorescence of *MP*^RSV^-GFP and synonymous mutant *MP*^RSV^SV-GFP, with and without expression of amiR MP observed at 48 hpi. Bars, 100 μm. (**B**,**C**) Western blot and Northern blot analysis of GFP-fused target mRNA from coinoculated *N. benthamiana* leaves. Loading control and rRNA were used to show that equal amounts of total protein and RNA were loaded. (**D**,**E**) Western blot and Northern blot analysis of GFP-unfused target mRNA from coinoculated *N. benthamiana* leaves. Loading control and rRNA were used to show that equal amounts of total protein and RNA were loaded. (**F**) qRT-PCR analysis of unfused target mRNA from coinoculated *N. benthamiana* leaves. Three independent experiments were performed for analysis. Asterisks mark significant differences according to *t*-test; ** *p* value ≤ 0.01.

**Figure 3 biology-11-00332-f003:**
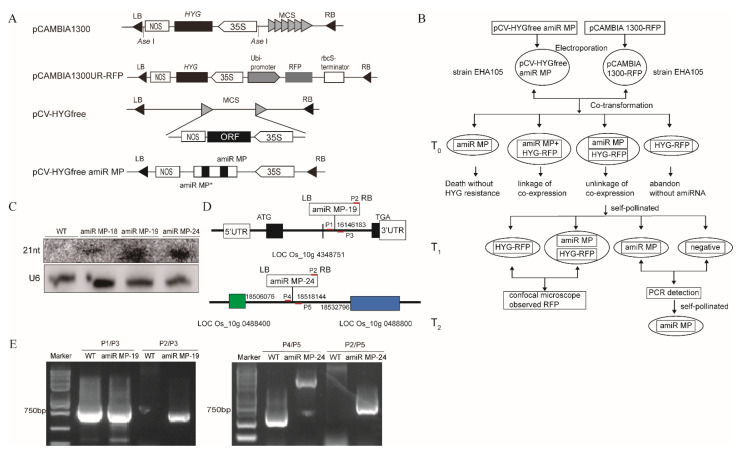
Generation of marker-free transgenic rice plants expressing amiR MP. (**A**) The construction of expression vectors pCAMBIA1300-RFP and pCV-HYGfree:amiR MP for cotransformation. The Ubi-P ubiquitin promoter and Nos-T nopaline synthetase terminator were used in this construction. (**B**) Schematic diagram showing the generation of marker-free transgenic rice expressing amiR MP. (**C**) Northern blot analysis of specific amiRNAs from transgenic plants amiR MP-18, amiR MP-19 and amiR MP-24. WT indicates wild-type plants. (**D**) Detection of insertion sites of amiR MP in transgenic lines by thermal asymmetric interlaced polymerase chain reaction (TAIL-PCR). ATG presents the starting codon, TGA presents termination codon, P1/P2/P3/P4/P5 the PCR primer sites. (**E**) Results of PCR reactions to identify the homozygous lines.

**Figure 4 biology-11-00332-f004:**
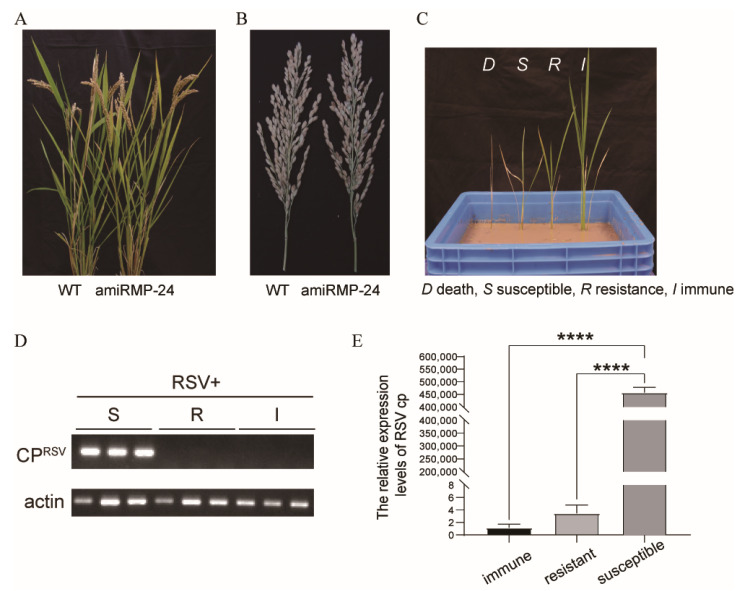
Transgenic rice plants showed high resistance against RSV infection. (**A**) The phenotypes of T_2_-generation transgenic amiR MP-24 plants compared with WT (Wild-type, Zhegeng-88) controls. (**B**) Spikes of amiR MP-24 and WT plants. (**C**) Symptoms in transgenic plants at 28 dpi following inoculation with RSV. *D* death: represents dead plants, *S* susceptible: stunting with chlorotic stripes, *R* resistant: mild stunting without chlorotic stripes, *I* immune: no symptoms. (**D**) RT-PCR detection of the accumulation of CP in RSV-infected plants with different symptoms. (**E**) RT-qPCR detection of the accumulation of CP in RSV-infected plants with different symptoms. Asterisks mark significant differences according to t-test; **** *p* value ≤ 0.0001.

**Table 1 biology-11-00332-t001:** Response of the T_2_-generation transgenic plants to inoculation with RSV. In total, 30 plants of each line were inoculated.

Transgenic Line	Numbers of Plants with Different Resistance Levels				Resistance Ratio (%) ^a^	Resistance Status ^b^
	I	R	S	D		
amiR MP-7	16	6	6	2	73	+
amiR MP-8	21	3	4	2	80	+
amiR MP-10	19	1	8	2	67	−
amiR MP-12	18	6	3	3	80	+
amiR MP-18	18	8	3	1	87	++
amiR MP-19	17	9	2	2	87	++
amiR MP-24	26	1	2	1	90	++
WT	6	5	18	1	37	−

*I* immune, *R* resistant, *S* susceptible, *D* dead; ^a^ resistant plants include I and R plants; ^b^ highly resistant ++; resistant +; susceptible −.

## Data Availability

Data are contained within the article or Appendix A.

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
