# Peer review of "Transgenic Rice Plants Expressing Artificial miRNA Targeting the Rice Stripe Virus *MP* Gene Are Highly Resistant to the Virus"

_biology, 2022, doi:10.3390/biology11020332_

Round 1

Reviewer 1 Report

The manuscript "Transgenic rice plants expressing artificial miRNA targeting ......................" describes the development of miRNA-based resistance rice lines. The manuscript is well designed and most cases are technically and scientifically sound. However, the evaluation of resistance of the transgenic lines after virus infection is not adequately presented. The results presented in Table 1 and the way the percentage of resistance is supposedly assessed are not really scientifically sound. The data presented in this table should be supplemented with the RT-qPCR data of the virus titre of individual transgenic lines and plants.

Reviewer 2 Report

RSV is one of the most devastating viral diseases in the world. Previously, the control effect of transgenic rice on RSV was not good. In this paper, the authors obtained marker-free transgenic rice expressing amiR-MP and highly resistant to RSV (>85%). The result provide a new germplasm resource to control RSV in the future. Overall this manuscript is well written and can be accepted for publication.

Author Response

Thank you very much for your comments.

Reviewer 3 Report

This manuscript reports authors’ work on the production and characterization of transgenic rice plants that express artificial miRNA (amiRNA) and are resistant to the rice stripe virus disease. In general, this manuscript contains useful information based on solid research results that can be used and referred by Biology’s readers. Given that the technology is valuable for the genetic improvement of crop varieties for the resistance to biotic and abiotic stresses, I attempt to recommend accepting this manuscript for publication in Biology. However, the organization of this manuscript and presentation of this manuscript is rather poor. Therefore, this manuscript needs substantial improvement. A few weak points are listed below for the authors to consider when revise the manuscript. Introduction The Introduction section is too short to clearly point out the existing key problems/challenges in fighting with the rice stripe virus disease and the advantages of using the amiRNA technology to solve these unsolvable problems. Therefore, the authors need to expand a little more to clearly introduce the background and adjectives for carrying out this research. Materials and methods In general, the “Materials and methods” section is too simple for readers to follow, more detailed information should be provided for readers to better understand how the authors conducted transgenic their work. For example: In the “Sequence and amiRNA prediction” section, the authors should provide some more detailed information about the sequence and construction of the amiRNA for readers to better understand the methodology. More information about the rice parental variety Zhegeng-88 used in the transformation should be provided as baseline information. Such as, is this a commonly cultivated rice variety or just a breeding line? Results Some parts of the “Results” section reads like “Materials and methods”. For example, some contents in the “Construction of the vector expressing amiRNA targeting RSV MP mRNA (amiR-MP)” section read like methods. Results should only contain what you obtained from the research. This statement “As shown in table S2 and figure 4A and B, the phenotypes between transgenic plants 199 and wild type were no significant difference. And the normal growth of transgenic plants 200 indicated that the expression of amiR MP in vivo had little effect on critical genes for 201 growth.” is not solid and cannot be accepted. In Table S2, no statistic analysis is provided. How could the authors get the above conclusion (no significant difference) without statistic support? Also, figure 4A and B (two photos) cannot support the statement “transgenic plants 199 and wild type were no significant difference”. Statistic analysis with necessary information should be provided. The pedigrees/generations of examined transgenic rice lines are not clearly indicated for each of the studies such as for the “Expression of amiR-MP decreased accumulation of MP mRNA”, “Generation of marker-free transgenic rice”, and “resistant to RSV by transgenic rice plants”. Therefore, the authors need to indicate which generations (T0, or T1, or Tn) of the transgenic plants were used for the specific study. Discussion Transgenic plants without antibiotic selection markers (marker free transgenic plants) are basic requirement if the product is aiming for commercial production. Therefore, the co-transformation becomes a common practice to produce transgenic plants. What are the advantages of the co-transformation techniques described in this manuscript? If yes, the authors need to discuss a little more on this aspect. As a transgenic product that is aimed for commercial production, the authors may need to provide some information about the potential biosafety (food or environmental safety) issues of the used transgenic technology and the obtained products. The manuscript will get improved by adding such information about the biosafety and possible solutions from professional and technique point of views. Such information will very useful for the readers. In addition, the English language of the manuscript needs substantial improvement. Some typo-errors were also found in the manuscript, for example, “exhibited a very much higher level of resistance” should be “exhibited a much higher level of resistance”, “the green fluorescence of the MPRSVSV-GFP and MPRSV-GFP were similar without transient…” should be “the green fluorescence of the MPRSVSV-GFP and MPRSV-GFP was similar without transient…”. The authors need to check carefully for such errors throughout the manuscript.

Round 2

Reviewer 1 Report

The manuscript has by no means been extensively revised. I still lack sufficient data on plant resistance. I suggested performing RT-qPCR for the evaluation of resistance presented in table 1, which is not fulfilled. Actually, besides the technical aspect, the resistance status of the transgenic lines should be the most important outcome of the manuscript.

Author Response

Response 1: Many thanks for the suggestions, and we totally agereed that identification of resistant transgenic lines in our study was one of the pivotal outcome in this paper. To evaluate the resistance of the plants, we here monitored the different symptom of plants (Fig. 4C) and added the RT-qPCR results of virus accumulation (added as Fig. 4E in the revised version) to confirmed the symptom. Actually, in the virus challenging assay, plants more often show different symptom, but not a single certain symptom. So we used the method reported before to evaluate the resistance of plants with different symptom. In the revised version, we added the RT-qPCR results of virus accumulation to confirm the symptom in the analysis. Indeed, we agreed that it would be better if we could detect virus accumulation in each plants of transgenic lines. But we are very sorry now for that we did not collect the samples of each plants for analysis. We have sown the plants, but the results cannot be added in this version on time. We will keep this constructive suggestion in our mind in the next work to evaluate the resistance of transgenic plants.

Reviewer 3 Report

The revision is generally acceptable, from which the MS showed improvement. However, the safety issue is a key for the technology that cannot be circumvented. We did not require the authors to provide results of biosafety assessments, which is beyond the MS per se, but some discussions predicting its potential safety issues based on the published results can be provided and discussed, as the authors argued in the response to reviewer’s comments (Because the transgenic rice plants express only a short 21nt sequence, researchers generally consider that the potential biosafety risks are likely to be minimal (Niu, et al. nature Biotechnology, 2006,24(11):1420-1428).) Some information from the relevant literature should be provided and discussed not only for experts but also for publics in the Discussion section.

Author Response

Response 1: Thanks for the suggestion, Content was added and highlighted in the discussion (highlighted in text, line 277-280):

  As a large number emergency of virus-resistance transgenic crops, the biological safety of transgenic plants raised concern, including off-target effects [44-45]. However, since the transgenic plants express only a short sequence through amiRNA-mediated strategies, these potential risks are likely to be minimal [11].